# *Brca1* Is Regulated by the Transcription Factor *Gata3*, and Its Silencing Promotes Neural Differentiation in Retinal Neurons

**DOI:** 10.3390/ijms232213860

**Published:** 2022-11-10

**Authors:** Jiejie Zhuang, Pei Chen, Yihui Wu, Qian Luo, Qiyun Wang, Shuilian Chen, Xi Chen, Zihua Jiang, Jin Qiu, Yan Li, Zhaohui Yuan, Jing Zhuang

**Affiliations:** Guangdong Provincial Key Laboratory of Ophthalmology and Visual Science, State Key Laboratory of Ophthalmology, Zhongshan Ophthalmic Center, Sun Yat-sen University, Guangzhou 510060, China

**Keywords:** GATA binding protein 3, *Brca1*, regulatory mechanism, neuronal differentiation

## Abstract

Previous studies have indicated that *Brca1* (*Breast cancer suppressor gene 1*) plays an important role in neural development and degenerative diseases. However, the bioactivity and regulatory mechanism of *Brca1* expression in retinal neurocytes remain unclear. In the present study, our data indicated that *Brca1* maintains the state of neuronal precursor cells. *Brca1* silencing induces differentiation in 661W cells. *Nestin*, a marker of precursor cells, was significantly decreased in parallel with *Brca1* silencing in 661W cells, whereas *Map2* (*Microtubule associated protein 2*), a marker of differentiated neurons, was significantly increased. Neurite outgrowth was increased by ~4.0-fold in *Brca1*-silenced cells. Moreover, DNA affinity purification assays and ChIP assays demonstrated that *Gata3* (*GATA binding protein 3*) regulates *Brca1* transcription in 661W cells. Silencing or overexpressing *Gata3* could significantly regulate the expression of *Brca1* and affect its promoter inducibility. Furthermore, the expression of *Gata3* generally occurred in parallel with that of *Brca1* in developing mouse retinas. Both *Gata3* and *Brca1* are expressed in the neonatal mouse retina but are developmentally silenced with age. Exogenous *Gata3* significantly inhibited neural activity by decreasing synaptophysin and neurite outgrowth. Thus, this study demonstrated that *Brca1* is transcriptionally regulated by *Gata3*. *Brca1*/*Gata3* silencing is involved in neuronal differentiation and maturation.

## 1. Introduction

*Brca1*, the breast cancer susceptibility gene, contains an N-terminal RING domain, nuclear localization signals (NLS) and two C-terminal BRCT domains [1]. A growing body of literature has demonstrated that *Brca1* functions in tumor suppression via DNA repair, genomic stability, cell cycle and transcriptional regulation during tumorigenesis and development [2,3,4]. Patients with mutations in *Brca1* are prone to developing breast and ovarian cancer [5,6]. Thus, *Brca1* has been identified as a target gene in precision medicine [7]. Moreover, the transcriptional mechanism of *Brca1* is well defined in tumor cells. However, the bioactivity and regulatory mechanism of *Brca1* expression in the retina remain unclear.

Although studies of *Brca1* in neural science are limited, much evidence indicates that *Brca1* may play an important role in neural development. Based on an analysis of in silico network construction in 2008, Bromberg et al. indicated that *Brca1* functions as a “hanger” in neural development [8]. The expression of *Brca1* declines with developmental stage progression [9]. Moreover, conditional knockout of Brca1 with a neural progenitor-specific driver could induce early apoptosis and reduce brain volume [10]. However, *Brca1* bioactivity is self-contradictory in neurodegenerative diseases. For example, *Brca1* can rescue neurons from cerebral ischemia/reperfusion injury [11]. In contrast, under pathological conditions, *Brca1* is abnormally colocalized with Tau in neurofibrillary tangles, a hallmark lesion of Alzheimer’s disease. Thus, these studies suggested that overexpression of *Brca1* was associated with proapoptotic phenotypes and ultimately resulted in neuron death [12,13]. Moreover, Evans et al. suggested that *Brca1* might initiate re-entry into the cell cycle of neurons [12]. Thus, *Brca1* may play various roles in different physiological conditions and stages. Its exact role in the retina still requires investigation.

The transcriptional mechanism of *Brca1* has also been well documented in tumor tissues. *Brca1* is regulated by methylation (DNA, histone), sumoylation or hypoxic conditions. The methylation status of the white blood cell *Brca1* promoter could be a predictive biomarker of ovarian cancer [14]. Sumoylation by SUMO1 negatively regulates *Brca1* transcription via modulation of promoter occupancy [15]. Conditions mimicking the tumor microenvironment, such as hypoxia, also silenced *Brca1* expression [16,17]. However, the mechanism of *Brca1* transcription in retinal neurocytes is not well defined. Our previous studies have demonstrated that the DNA demethylation agent 5-Aza could upregulate *Brca1* expression in postnatal rat retinal neurons [18]. However, we did not find a methylated site in the *Brca1* promoter through bisulfite sequencing. Thus, more investigation is required to reveal the transcriptional mechanism of *Brca1* expression in retinal neurocytes.

To address these questions, we investigated *Brca1* bioactivity during proliferation, differentiation and neurite outgrowth using cell lines and primary retinal cells coupled with gene interference. Transcription factors regulating *Brca1* expression were identified using DNA affinity purification, ChIP and promoter inducibility assays [19]. Thus, this study reveals the bioactivity and transcriptional mechanism of *Brca1* expression in retinal neurocytes.

## 2. Results

### 2.1. Downregulation of Brca1 Promotes Differentiation of Retinal Precursor Cells

The 661W cell, a cone-photoreceptor-specific expressing precursor-like cell line, can be used as an ideal tool for investigating the differentiation of retinal neurocytes [20,21]. As shown in Figure 1A, 661W cells were characterized as having characteristics of stem cells. *Nestin* (red), a marker of precursor cells, was expressed in the cytoplasm, and *Brca1* was expressed in the nuclei of 661W cells; there was no obvious staining for *Map2*, a marker of neurons.

To reveal its bioactivity in retinal progenitor cells, *Brca1* was silenced by siRNA interference. After 48 h of treatment with si*Brca1* or siControl, total protein was extracted from 661W cells and measured by Western blot. As shown in Figure 1B,C, *Brca1* silencing (*Brca1*: siControl, 1; si*Brca1* 0.29 ± 0.15; * *p* < 0.05) significantly decreased the expression of *Nestin* in 661W cells (*Nestin*: siControl, 1; si*Brca1* 0.627 ± 0.05; ** *p* < 0.01), which implies that *Brca1* is upstream of *Nestin*. *Map2* can be alternatively spliced into multiple isoforms in neural cells. The high-molecular-weight isoform *Map2* A/B functions in neuronal maturation, and the low-molecular-weight isoform *Map2* C/D promotes neurogenesis and neurite initiation. As shown in Figure 1D,E, the expression of *Map2* (C/D) was significantly increased in *Brca1*-silenced 661W cells compared with control cells (C/D: siControl, 1; si*Brca1*, 1.79 ± 0.19; * *p* < 0.05), which indicates downregulation of *Brca1* mainly promotes neural differentiation but not maturation.

To observe changes in cell shape following treatment with si*Brca1*, the cells were double stained with anti-*Map2* and anti-*Nestin* antibodies. As shown in Figure 2A, *Map2* (green) was expressed in 661W cells treated with si*Brca1*. Fully *Map2*-positive cells had a smaller soma and longer dendrites (green, white arrow). In contrast, little *Nestin* expression was observed in differentiated cells with developed neurites (red, white arrowhead). Further experiments confirmed these results via double staining with anti-*Brca1* and anti-*Map2*/anti-*Nestin* in *Brca1*-silenced cells. *Brca1* silencing significantly promoted retinal neurite outgrowth in 661W cells (siControl, 1.16 ± 0.11; si*Brca1*, 4.31 ± 0.14; ** *p* < 0.01) (Figure 2B). Accordingly, *Brca1* silencing significantly inhibited cell viability (siControl, 1; si*Brca1*, 0.65 ± 0.06; ** *p* < 0.01) (Figure 2C). Collectively, these data indicate that *Brca1* is involved in maintaining neural precursor status and that its silencing promotes neuronal differentiation.

### 2.2. Brca1 Is Transcriptionally Regulated by Gata3 In Vitro

To reveal the transcriptional mechanisms involved in regulating *Brca1* expression in retinal neurons, DNA affinity purification assays were performed to investigate the proteins that bind to the *Brca1* promoter. The experimental procedure is summarized in Figure 3A. Several bands (asterisks) appeared only in the presence of the *Brca1* promoter (+) and disappeared in the absence of the *Brca1* promoter (−) due to the higher NaCl elutions (Figure 3B). These bands containing proteins that bound the *Brca1* promoter with high affinity and specificity were digested and identified by mass spectrometry. The following proteins were identified: *Gata3*, Twist 1, ZFP521 and SFPQ, among others (Figure 3C). As reported previously, these proteins are known to be transcription regulators that bind to other promoters [22,23,24,25]. Among them, *Gata3* is characterized as an important transcription factor involved in central nervous system development [26]. Moreover, *Gata3* was strongly expressed and colocalized with *Brca1* in the nucleus of 661W cells (white arrows), as evidenced by immunofluorescence assays (Figure 3D).

To further explore the potential relationship between *Gata3* and *Brca1*, 661W cells were transfected with either *Gata3* siRNA (si*Gata3*) or the plasmid pFLAG-*Gata3*; control cells were transfected with either scramble siRNA (siControl) or the empty vector, pcDNA3.1. At 24 h after transfection, total protein was extracted and measured by Western blot assay. As shown in Figure 4A,B, *Gata3* expression was significantly downregulated by *Gata3* siRNA transfection compared with that of the control siRNA (siControl, 1; si*Gata3*, 0.33 ± 0.10; ** *p* < 0.01) (Figure 4B). Consistently, the expression of *Brca1* was also significantly decreased in *Gata3*-silenced 661W cells (siControl, 1; si*Gata3*, 0.57 ± 0.21; * *p* < 0.05) (Figure 4A,B). In contrast, the expression of *Brca1* was increased in parallel with the upregulation of *Gata3* in the 661W cells transfected with the pFLAG-*Gata3* plasmid (Figure 4C). The relative protein expression levels of *Gata3* and *Brca1* in 661W cells are presented as a histogram, demonstrating that *Gata3* expression was upregulated by 10.8-fold (10.80 ± 3.50, * *p* < 0.05) after pFLAG-*Gata3* plasmid transfection and, in parallel, *Brca1* expression was also increased by 1.57-fold (1.57 ± 0.14, ** *p* < 0.01) (Figure 4D). The inconsistency in the ratio of their increased expression implies that *Gata3* is a transcription factor that regulates *Brca1*.

To confirm whether *Gata3* directly regulates the transcription of *Brca1*, luciferase reporter and ChIP assays were used. *Gata3*-silenced or *Gata3*-overexpressing 661W cells were cotransfected with pCMV-RL-pFLAG-*Gata3* or pcDNA (control). The levels of luciferase activity were measured and normalized to Renilla luciferase. As shown in Figure 4E, the luciferase activity of the *Brca1* promoter decreased by approximately 60% with the downregulation of *Gata3* expression (siControl, 1; si*Gata3*, 0.41 ± 0.08; ** *p* < 0.01) and increased by approximately 22% with the upregulation of *Gata3* expression in 661W cells (Control, 1; pFLAG-*Gata3*, 1.22 ± 0.1; * *p* < 0.05). Moreover, the binding status of *Gata3* to the *Brca1* promoter in 661W cells was detected by ChIP assays using an antibody against *Gata3* and a control normal rabbit IgG antibody. Our data showed that, compared with mock precipitation, there was an approximately 2.2-fold greater enrichment for the *Brca1* promoter with the *Gata3* antibody (IgG, 1; anti-*Gata3*, 2.18 ± 0.44; * *p* < 0.05), suggesting that *Gata3* binds to the *Brca1* promoter in vitro (Figure 4F). Thus, this evidence strongly suggests that *Gata3* may bind to the *Brca1* promoter and regulate its activity, eventually altering the expression of the *Brca1* protein.

### 2.3. Gata3 Silencing Promoted Cell Differentiation Similar to Brca1 Silencing in 661W Cells

Since *Gata3* positively regulated the expression of *Brca1*, *Gata3* may also be involved in neuron differentiation. To verify this hypothesis, control and *Gata3*-silenced 661w cells were double stained with anti-*Nestin* and anti-*Map2*. As shown in Figure 5A, consistent with our previous study, *Nestin* was strongly expressed in control 661W cells (red), whereas the *Map2* signal was very weak (green) [27]. By contrast, weak *Nestin* and strong *Map2* staining were observed in *Gata3*-silenced 661W cells. In addition, unlike the control cells that appeared ovular and had elongated central axes with no distinct neuritis, the *Gata3*-silenced 661W cells presented neurite outgrowth and displayed primary and secondary branching (white arrowheads). *Nestin* was weakly expressed in these neurite outgrowths (white arrows). Similar to *Brca1* silencing, silencing *Gata3* significantly reduced cell viability (siControl, 1; si*Gata3*, 0.51 ± 0.05; ** *p* < 0.01) (Figure 5B). Silencing *Brca1* or *Gata3* induced synaptophysin expression in 661W cells (Figure 5C). This evidence indicates that the level of cell differentiation is correlated with the level of *Gata3* expression in 661W cells, further confirming the role of *Gata3* in cell differentiation.

### 2.4. The Expression of Gata3 in Developing Mouse Retinas Generally Occurs in Parallel with That of Brca1 in Precursor-like Cells

To further verify our in vitro results, we quantitatively examined the protein expression patterns of *Brca1* and *Gata3* in the mouse retina by Western blot. The relative intensities of the bands obtained by Western blot were quantified by densitometry and normalized to Tubulin levels. As shown in Figure 6A,B, the expression of both *Brca1* and *Gata3* was significantly induced after birth, with intense bands detected in the lysates of P1 mouse retinas; however, very weak expression was observed in adult mouse retinas (*Brca1*: P1, 0.316 ± 0.187; adult: 0.041 ± 0.007; * *p* < 0.05; *Gata3*: P1, 0.848 ± 0.272; adult, 0.293 ± 0.02; ** *p* < 0.070) (Figure 6A,B).

Moreover, the localization profiles of *Brca1* and *Gata3* in the mouse retina were analyzed by immunohistochemistry and immunofluorescence (Figure 6C. *Brca1* was predominantly distributed in the ganglion layer (GCL) and inner nuclear layer (INL) of the immature P1 neonatal mouse retina. Slight immunoreactive cells were observed in the GCL and INL of the differentiated retina neurons of adult mice. The expression and localization profiles of *Gata3* were coincident with those of *Brca1* in the mouse retina. Taken together, this evidence reveals similar expression patterns between *Nestin*, *Brca1* and *Gata3* in the mouse retina, further suggesting a regulatory relationship in the differentiation and maturation of retinal neurons.

Primary retinal neurons (P1) were double stained with anti-*Map2* and anti-*Gata3*/anti-*Brca1*. Figure 6D shows that *Gata3* and *Brca1* were expressed in P1 retinal neurons. *Map2* (green) was located in neural outgrowths and the cytoplasm, whereas *Gata3* and *Brca1* (red) were located in the nuclei of retinal neurons.

### 2.5. Brca1 Might Be Transcriptionally Regulated by Gata3 in Primary Retinal Neurocytes In Vitro

Since the *Brca1*/*Gata3* pathway is involved in the differentiation of retinal cell lines, we further confirmed its role in primary retinal neurocytes. Primary retinal neurocytes (P1) were infected with AAV-Re-*Gata3*-GFP or AAV-Re-GFP adenovirus. At 5 days after infection, GFP was highly expressed in the cells (Figure 7A). Western blot assays indicated that a more than 60-fold increase in *Gata3* expression was observed in primary retinal cells infected with AAV-Re-*Gata3*-GFP compared with that of cells infected with AAV-Re-GFP (AAV-Re-GFP, 1; AAV-Re-*Gata3*-GFP, 61.76 ± 10.65; ** *p* = 0.01) (Figure 7B,C). In parallel, *Brca1* expression was mildly increased (AAV-Re-GFP, 1; AAV-Re-*Gata3*-eGFP, 1.35 ± 0.10; * *p* < 0.05) (Figure 7C). Thus, these data further indicate that *Gata3* might be a transcription factor that regulates the expression of *Brca1*.

To confirm the impact of *Gata3* overexpressing on retinal neurocytes, staining with anti-*Map2* in primary retinal neurocytes exposed to AAV infection was performed. As shown in Figure 7C, primary retinal neurocytes infected with AAV-Re-GFP exhibit strong *Map2* expression and normal neurite growth. In contrast, AAV-Re-*Gata3*-GFP infection induces weak *Map2* expression and limited neurite growth in GFP-positive neurocytes, while the GFP-negative neurons show strong *Map2* staining and distinct neuritis (White arrow). Our data indicate that *Gata3*-overexpressing impedes differentiation of retinal neurocytes.

Moreover, we found that overexpressing *Gata3* also induced cell death of primary retinal neurons. The cells were analyzed at 5 days after transfection by TUNEL assays. As shown in Figure 7D,E, overexpressing *Gata3* induced a significant increase in death (AAV-Re-GFP, 17.11 ± 1.26%; AAV-Re-*Gata3*-GFP, 35.10 ± 3.22%; ** *p* < 0.01). The CCK-8 assay also indicated that exogenous *Gata3* decreased cell viability (AAV-Re-GFP, 1; AAV-Re-*Gata3*-GFP, 0.344 ± 0.16; * *p* < 0.05) (Figure 7F).

## 3. Discussion

In this study, we first demonstrated that *Brca1* is involved in maintaining the state of retinal precursor cells. Silencing *Brca1* by siRNA interference significantly downregulated *Nestin*, a marker of stem cells, increased the expression of *Map2*, and promoted neurite outgrowth in 661W cells (Figure 1 and Figure 2). Moreover, the expression pattern of *Brca1* in the retina in vivo was similar to that in vitro (Figure 6). *Nestin* was highly expressed in the retina of P1 (Figure 6), which indicated that the cells of this stage have characteristics of neural precursor cells. *Brca1* was expressed in postnatal mouse retinas but not adult retinas. Thus, these data indicated that *Brca1* silencing is involved in the differentiation of retinal precursor cells. This conclusion is partially consistent with previous studies. For example, *Brca1* was expressed at leading edges in migrating cells, particularly in proliferative zones, during brain development [28,29]. *Brca1* plays a role as a centrosomal factor in maintaining neural progenitors [13]. Oligodendrocyte precursor cells (OPCs) were also maintained by *Brca1* and Brm [30]. Inactivation of *Brca1* by cisplatin could increase growth cones in embryonic mouse brain precursor cells [31].

However, why does silencing *Brca1* cause such typical neural differentiation? We speculate that *Brca1* might function as a transcription factor in precursor cells. A previous study demonstrated that Sox2 expression depended on *Brca1* via chromatin remodeling and histone modification in neural stem-like cells. *Brca1* contributes to the conversion of oligodendrocyte precursors into neural stem cells [30]. Moreover, *Brca1* might be a candidate ubiquitin ligase to degrade proteins in cholinergic neuritis [28]. However, for retinal precursor cells, more investigation is required to reveal the exact mechanism of *Brca1*-mediated differentiation in the future.

Second, a strength of our study was that *Gata3* transcriptionally regulated *Brca1* in retinal cells. Analysis of DNA affinity purification and ChIP assays indicated that *Gata3* binds to the *Brca1* promoter in 661W cells (Figure 3). Silencing *Gata3* also decreased promoter inducibility and the expression level of *Brca1* in vitro. Previous studies partially support our discovery. Celikkaya et al. reported that *Gata3* promoted the neural progenitor state [26]. *Gata3* is required for reactive proliferation of progenitors [32]. However, the decreased expression of *Brca1* was not consistent with that of *Gata3* (Figure 4A,B). Accordingly, exogenous *Gata3* expression was also up to ~10-fold higher compared to the control, while *Brca1* was only increased by ~50% (Figure 4D). Moreover, exogenous *Gata3* notably induced an upregulation of *Brca1* in primary postnatal retinal neurons (Figure 7A,B,D). However, the increased *Brca1* expression was notably lower than that of *Gata3*. Therefore, we speculated that *Gata3* might act as one of the transcription factors promoting *Brca1* promoter activity along with other factors in postnatal retinal cells.

In addition, our data indicated that the effect of *Brca1* on cell viability depends on cell type. Currently, there are different viewpoints about *Brca1* in the central nervous system. Some studies suggested that the function of *Brca1* was unique in neurons and increasing *Brca1* could promote cell cycle re-entry and induce neuronal death in Alzheimer’s diseases [12,13]. However, Xu et al. indicated that *Brca1* protected neurons from cerebral ischemia/reperfusion injury through the NRF2-mediated antioxidant pathway [11]. Our previous study also indicated that *Brca1* contributed to neural viability by enhancing DNA stability in RGC5, an immortal neural cell line [33]. Why did these studies suggest different conclusions? The data presented here can account for this discrepancy. We found that *Brca1* and *Gata3* promoted the viability of precursor cells (Figure 1F and Figure 5D), whereas cell viability was decreased in differentiated P1 retinal neurons that re-expressed both genes (Figure 7). Thus, we speculate that *Brca1* rescues neurons by affecting precursor cells in cerebral tissue in cerebral ischemia/reperfusion injury. *Brca1* and *Gata3* play different roles in neural progenitor cells and fully differentiated neurons.

## 4. Materials and Methods

661W cell culture. The mouse retinal cell line, 661W, purchased from ATCC (Manassas, VA, USA), was cultured in Dulbecco’s modified Eagle’s medium (DMEM, Gibco, Carlsbad, CA, USA) supplemented with 10% fetal bovine serum (FBS; Gibco, Carlsbad, CA, USA) and 1% penicillin/streptomycin (Gibco, Carlsbad, CA, USA) in a humidified 5% CO_2_ incubator.

Primary mouse retinal neuron culture and treatment. Primary mouse retinal neurons were cultured as described previously. Mice were provided by the animal center of Zhongshan Ophthalmic Center, Sun Yat-sen University, China. The culturing protocol followed Reference [34]. The cells were maintained in complete medium (DMEM supplemented with 10% FBS) and characterized by *Map2* staining (Boster, Wuhan, China, BM1243, 1:100). Three days after culture, the cells were used in the experiments described below.

Reporter and plasmid construction. The PGL3-WT plasmid carrying the mouse *Brca1* promoter region (−1012 to +208) was generated as follows. The required *Brca1* promoter region was amplified by PCR using the primer pair 5′-GGGGTACCCCCTTCCTTACCAGCTTTCCGC-3′ and 5′-CCCAAGCTTGGGCTGTTCCTCAGGGCTGTCTC-3′ (Kpn I and Hind III sites are underlined, respectively). The PCR products and the PGL3-WT vector were digested with Kpn I and Hind III, and the *Brca1* promoter fragment (1220 bp) was inserted into PGL3-WT at identical sites following the manufacturer’s protocol, yielding the vector PGL3-*Brca1*-promoter. The integrity of the fragment was verified by DNA sequencing.

Virus preparation. The recombinant adeno-associated viruses AAV-Re-GFP and AAV-Re-*Gata3*-GFP were produced and packaged into recombinant adeno-associated virus 2 retro [35] by Cyagen Biosciences using the mouse Gata binding protein 3 (*Gata3*) cDNA sequence (accession number NM_001002295.2). AAVs were stored in 150 mM NaCl, 2 mM MgCl2 and 50 mM Tris (pH 8.0) at −80 °C for less than 1 year and thawed on ice on the day of use.

*Brca1* promoter-reporter assay. Mouse retinal 661W cells were seeded in 24-well plates and cultured overnight. Then, the luciferase reporter pGL3-*Brca1*-luc was cotransfected with different plasmids or siRNAs, including pcDNA, a *Gata3* overexpression vector (pFLAG-*Gata3*; Addgene, Cambridge, MA, USA), and *Gata3* siRNA and scramble control siRNA. The pCMV-RL plasmid encoding Renilla luciferase was included in all the samples to monitor transfection efficiency. Twenty-four hours after transfection, the cells were harvested, and luciferase activity was measured using the Dual-Glo Luciferase Assay (Promega, Madison, WI, USA). The levels of firefly luciferase activity were normalized to Renilla luciferase activity.

Real-time RT–PCR. Total RNA was isolated with TRIzol Reagent (Invitrogen, Carlsbad, California, USA). The following primer pairs were used: mouse *Gata3*, 5′-CGAGATGGTACCGGGCACTA-3′ (sense) and 5′-GACAGTTCGCGCAGGATGT-3′ (antisense); mouse *Brca1*, 5′-TCTGGCAGCATGTTCTCTTC-3′ (sense) and 5′-CTCATTCCCACACTGGTGAC-3′ (antisense); mouse β-actin, 5′-AGGTCATCACTATTGGCAACG-3′ (sense) and 5′-ACGGATGTCAACGTCACACTT-3′ (antisense); mouse *Map2*, 5′-AAGTCACTGATGGAATAAGC-3′ (sense) and 5′-CTCTGCGAATTGGTTCTG-3′ (antisense); mouse *Nestin*, 5′-TCAAGATGTCCCTTAGTCTGGA-3′ (sense) and 5′-TGGTCCTCTGGTATCCCAAGG-3′ (antisense).

Western blotting. Total protein was extracted from cells using RIPA buffer supplemented with PMSF. The following primary antibodies were used: *Gata3* (Abcam, Cambridge, MA, USA, ab199428, 1:500), *Brca1* (Santa Cruz, Dallas, TX, USA, ab199428, 1:500), *Map2* (Boster, Wuhan, China, BM1243, 1:1000), *Nestin* (Millipore, Burlington, MA, USA, mab353, 1:1000) and synaptophysin (Abcam, USA, mab353, 1:1000). The membrane was incubated with horseradish peroxidase-conjugated secondary anti-rabbit (CST, USA, 7074s, 1:10,000) or anti-mouse antibody (CST, USA, 7076s, 1:10,000). Gadph (Protein Tech Group, Wuhan, China, 10494-1-AP, 1:2000) served as a loading control. Protein bands were detected using an enhanced chemiluminescence detection system (Millipore, USA).

Immunofluorescence analysis. Immunofluorescence assays were performed according to the standard protocol. In brief, cells were fixed with paraformaldehyde after the treatments mentioned above. The primary antibodies included mouse anti-*Map2* (Boster, China, BM1243, 1:100), rabbit anti-*Map2* (Abcam, USA, ab254264, 1:200), rabbit anti-*Nestin* (Millipore, USA, mab353, 1:100), mouse anti-synaptophysin (Abcam, USA, ab8049, 1:500), rabbit anti-*Gata3* (Abcam, USA, ab199428, 1:100), rabbit anti-*Brca1* (Proteintech Group, Wuhan, China, 20649-1-AP) and mouse anti-Flag (Sigma, St. Louis, MO, USA, ab199428, 1:100). Cell death was detected by the TUNEL staining (kit from Elabscience, China, E-CK-A320) according to the manufacturer’s manual, the GFP+TUNEL+ cells were calculated as dead cells.

DNA affinity purification. DNA affinity purification was performed following previously described procedures with some modifications [36]. The nuclear extract was isolated from 661W cells using a nuclear and cytoplasmic protein extraction kit (Beyotime, China, P0027). The *Brca1* promoter corresponding to positions −28 to −407 relative to the transcription initiation site (+1) was amplified by PCR using 5′-biotin-labeled primers (up: 5′-biotin-CAGGAAGGCTGAGGGAGGGA-3′; down: 5′-CTAAAATTCCCGCGCTCTCC-3′) and bound to streptavidin Dynabeads (Invitrogen, Carlsbad, CA, USA, 65305). Nuclear extracts were incubated with the DNA beads and washed extensively with BS/THES buffer, taking advantage of the magnetic properties of the Dynabeads when changing buffers. As a negative control, similar incubation was also carried out with beads lacking DNA bait. Bound proteins were eluted by increasing NaCl concentrations to 200, 300, 500, 750 and 1000 mM. Lower NaCl concentrations (100–300 mM) eluted proteins that bound to the promoter through weak interactions, while elution with higher NaCl concentrations (500–1000 mM) yielded proteins that have high affinity for the *Brca1* promoter. Eluted samples were analyzed by SDS–PAGE followed by silver staining. Protein bands were excised, digested with trypsin and analyzed by liquid chromatography (LC)-MS/MS spectrometry on a Q Exactive (Thermo Fisher Scientific, San Jose, CA, USA) coupled online to high-performance liquid chromatography (HPLC). Proteins were identified by database searches using the Mascot search engine (Matrix Science, London, UK), and the data were collated into a list of proteins containing annotated transcription factors exclusively.

Immunohistochemical assay. Immunohistochemical assays were performed on retinal slides of postnatal day 1 and adult mice according to the manufacturer’s protocols of the SABC-POD (F) rabbit IgG kit (Boster, China, SA1028). Rabbit anti-*Gata3* (1:100, Abcam, USA, ab199428, 1:100) and rabbit anti-*Brca1* (Santa Cruz, USA, sc-135732, 1:100) were used as primary antibodies, and biotinylated anti-rabbit IgG antibodies were used as secondary antibodies. Following washing, the sections were developed with 3′-diaminobenzidine tetrahydrochloride (DAB) peroxidase substrate (Boster, China, AR1000) and counterstained with hematoxylin.

Cell viability assayed by CCK-8. Cell viability was determined by a Cell Counting Kit-8 (CCK-8) assay (Dojindo, Japan). Cells were incubated with CCK-8 reagent for 2 h at 37 °C followed by measuring the optical density at 450 nm. Cell viability was normalized to the untreated control.

ChIP assay. ChIP was carried out using a ChIP Assay Kit (Upstate Cell Signaling Solutions, Lake Placid, NY, USA) following the manufacturer’s instructions. Briefly, the chromatin from cells crosslinked with 1% formaldehyde was sheared by sonication. Some of the sheared chromatin was kept as an input control, and the rest was incubated with either *Gata3* antibody (Abcam, Cambridge, MA, USA, ab199428, 1:100) or normal IgG as a negative control. The precipitated DNA was subjected to real-time PCR using the following primers that span the *Gata3* binding site on the *Brca1* promoter: forward, 5′-ACCTCGTTTTGCAACTGCTT-3′; reverse, 5′-GGTTTGCGATTGGCTACCTA-3′.

RNA interference. The sequences of *Brca1* siRNA and control siRNA were as follows: *Brca1*-siRNA, 5′-GCAGGAGCCAAAUCUAUAA(dTdT)-3′ and control siRNA, 5′-CCUACGCCACCAAUUUCGU(dTdT)-3′. The siRNA targeting *Gata3* is a pool of three different sequences: *Gata3*-siRNA-1, 5′-GACGGAAGAGGUGGACGUA(dTdT)-3′; *Gata3*-siRNA-2, 5′-UCGUACAUGGAAGCUCAGU(dTdT)-3′; *Gata3*-siRNA-3, 5′-GAUUUCAGAUCUGGGCAAU(dTdT)-3′. The oligos were synthesized by RiboBio (Guangzhou, China). Transfections were performed with Lipofectamine RNAiMAX (Invitrogen Waltham, MA, USA), and the expression levels of *Gata3* and *Brca1* were measured by Western blot at 24 h post-transfection.

Statistical analysis. All in vitro experiments were performed at least 3 times. Data are expressed as the means ± SEMs. The differences between mean values were evaluated with the two-tailed Student’s *t*-test (for two groups), analysis of variance (ANOVA, for 2 groups) and analysis of variance (for >2 groups). All calculations and statistical tests were performed using the computer programs Microsoft Excel 2003 (Microsoft, Redmond, WA, USA) and SPSS 11.5 (SPSS, Chicago, IL, USA). *p* < 0.05 was considered significant for all analyses.

## 5. Conclusions

In conclusion, the study presented here sheds light on the underlying transcriptional mechanism and bioactivity of *Brca1* expression in retinal neurons. *Brca1* and the upstream *Gata3* are likely to be relevant for neuronal differentiation.

## Figures and Tables

**Figure 1 ijms-23-13860-f001:**
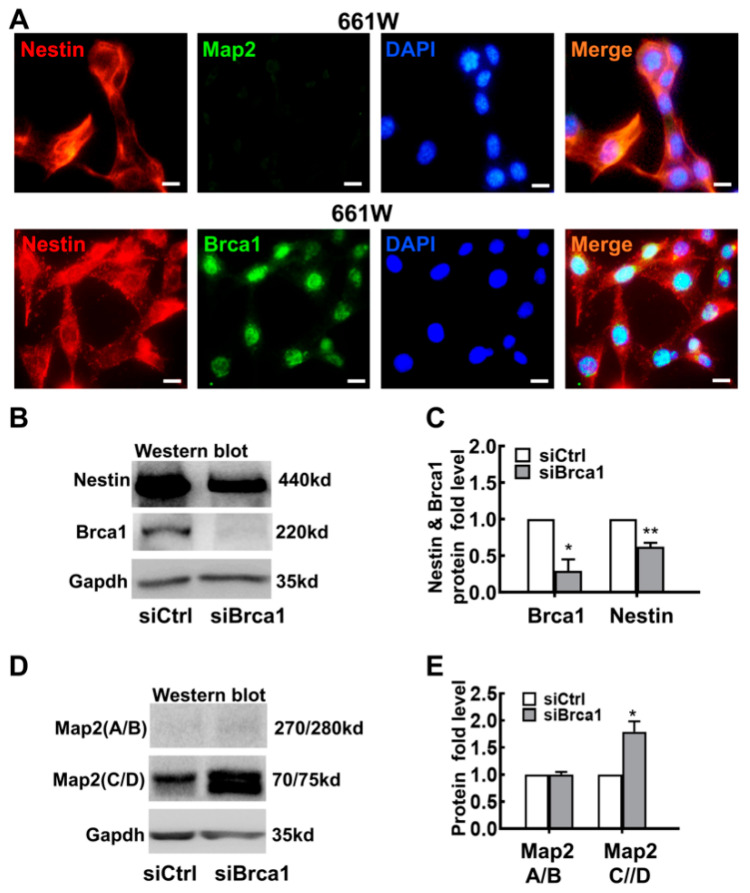
*Brca1* silencing induces differentiation of 661W cells. (**A**) Strong *Nestin* expression (red) and faint *Map2* expression were observed in the cell cytoplasm of 661W cells. *Brca1* was expressed at high levels in the cellular nuclei of 661W cells (green). The nuclei were stained with DAPI (blue). (**B**) *Nestin* expression was decreased in *Brca1*-silenced 661W cells, as evidenced by Western blot. (**C**) The relative expression levels of *Nestin* and *Brca1* in 661W cells are presented as a histogram. (**D**) *Brca1* silencing significantly increased the expression of *Map2* (**C**/**D**) in 661W cells. (**E**) The relative protein expression levels of *Map2* (**A**/**B**) and *Map2* (**C**/**D**) are presented as histograms. Scale bars represent 10 μm. The asterisks indicate statistically significant differences (* *p* < 0.05, ** *p* < 0.01).

**Figure 2 ijms-23-13860-f002:**
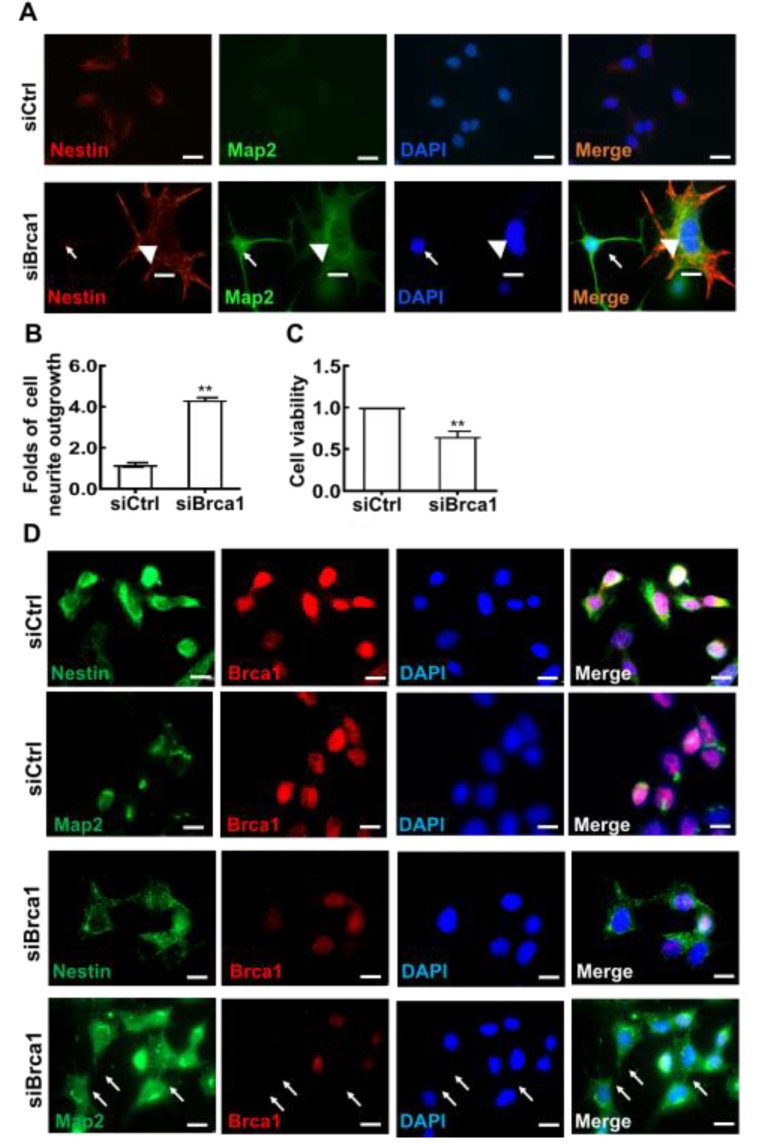
*Brca1* is involved in the differentiation of 661W cells. (**A**,**D**), *Brca1* interference decreased *Nestin* expression (red, cell cytoplasm) while increasing *Map2* expression in 661W cells (green, cell cytoplasm). The cell nuclei were stained with DAPI (blue). The white arrows indicate dendrites or neurites stained by *Map2*, the white arrowhead indicates neurite stained by *Nestin*. (**B**) *Brca1* silencing promoted retinal neurite outgrowth. Scale bars represent 20 μm. (**C**) *Brca1* interference by siRNA inhibited the viability of 661W cells. The asterisks indicate statistically significant differences (** *p* < 0.01).

**Figure 3 ijms-23-13860-f003:**
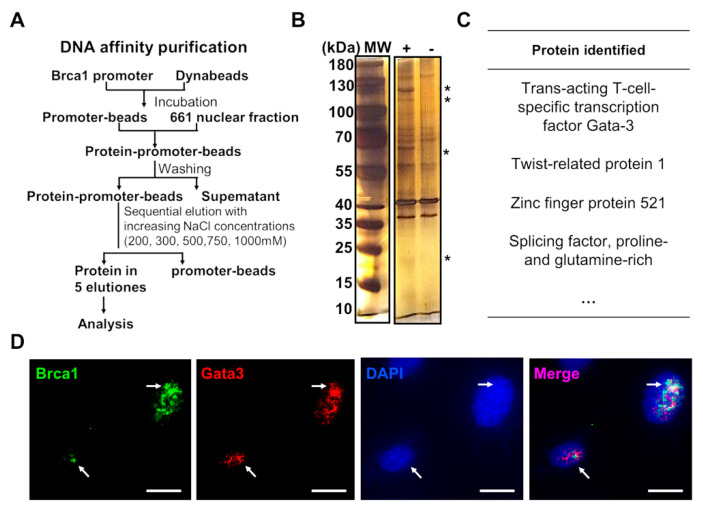
The transcription mechanisms of *Brca1* expression. (**A**) A summary of the experimental procedures used for DNA affinity purification assays. (**B**) Several bands specifically appeared (asterisk) in the presence of the *Brca1* promoter (+), but disappeared in the absence of the *Brca1* promoter (-). (**C**) Proteins that might be related to transcriptional mechanisms were identified by mass spectrometry. (**D**) *Gata3* (red) is strongly expressed and colocalized with *Brca1* (green) in the cellular nuclei of 661W cells. Scale bars represent 10 μm. The asterisks indicate statistically significant differences (**p* < 0.05, ** *p* < 0.01). The white aroows indicate colocalization of Brca1 and Gata3 in the nucleus.

**Figure 4 ijms-23-13860-f004:**
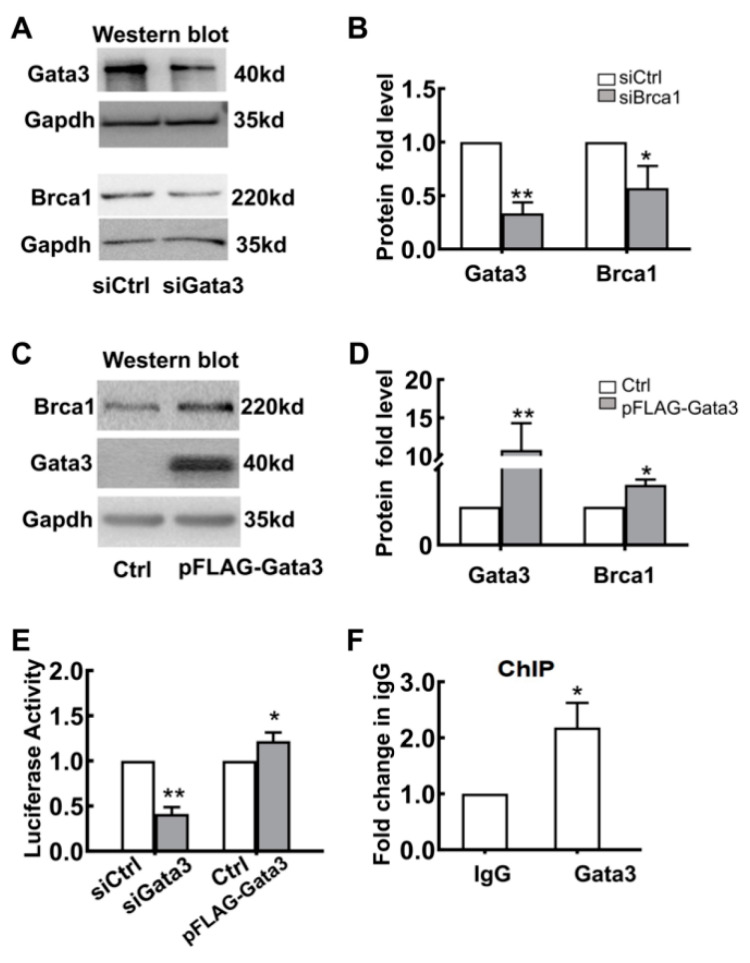
*Brca1* is transcriptionally regulated by *Gata3***.** (**A**) *Brca1* expression decreased in the *Gata3*-silenced 661W cells, as evidenced by Western blot. (**B**) The relative protein expression levels of *Gata3* and *Brca1* are presented as histograms. (**C**) Exogenous *Gata3* significantly increased *Brca1* expression in 661W cells. (**D**) The relative protein expression levels of *Gata3* and *Brca1* are presented as histograms. (**E**) The luciferase activity of the *Brca1* promoter decreased with the downregulation of *Gata3* but increased with the upregulation of *Gata3* in 661W cells. (**F**) *Gata3* might bind to the *Brca1* promoter and regulate its activity, as evidenced by ChIP assays. The asterisks indicate statistically significant differences (* *p* < 0.05, ** *p* < 0.01).

**Figure 5 ijms-23-13860-f005:**
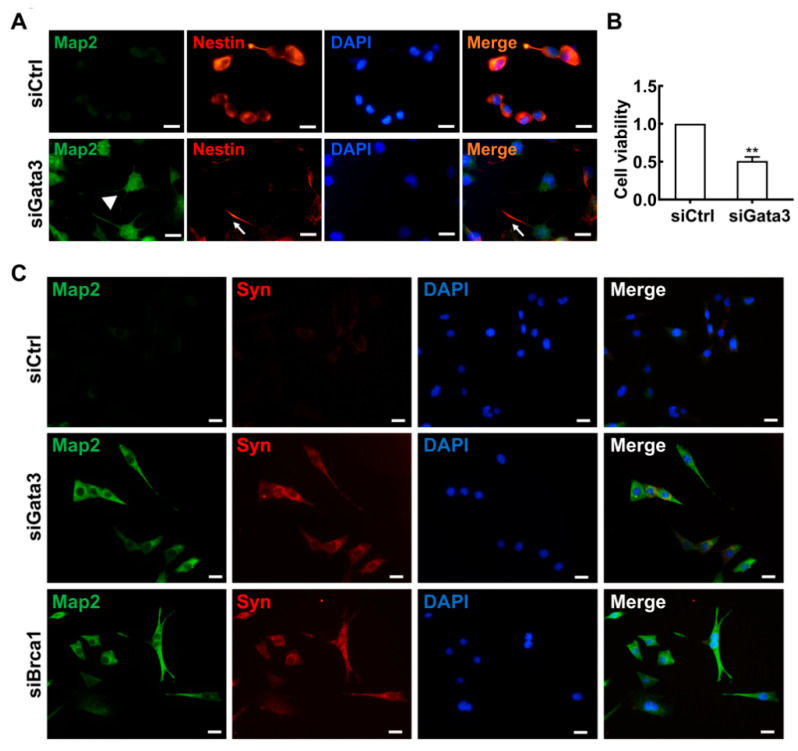
*Gata3* inhibition induces cell differentiation in 661W cells. (**A**) Strong *Nestin* (red) and weak *Map2* (green) expression were observed in the 661W cells. In *Gata3*-silenced 661W cells, *Nestin* expression was decreased, and *Map2* expression was observed. The white arrow indicates neurite outgrowth stained by *Nestin*, the white arrowhead indicates neurite outgrowth stained by *Map2*. (**B**) *Gata3* inhibition decreased the viability of 661W cells. (**C**), Both *Gata3* silencing and *Brca1* silencing induce synaptophysin (red, cell cytoplasm) expression in 661W cells. Scale bars represent 20 μm. The asterisks indicate statistically significant differences (** *p* < 0.01).

**Figure 6 ijms-23-13860-f006:**
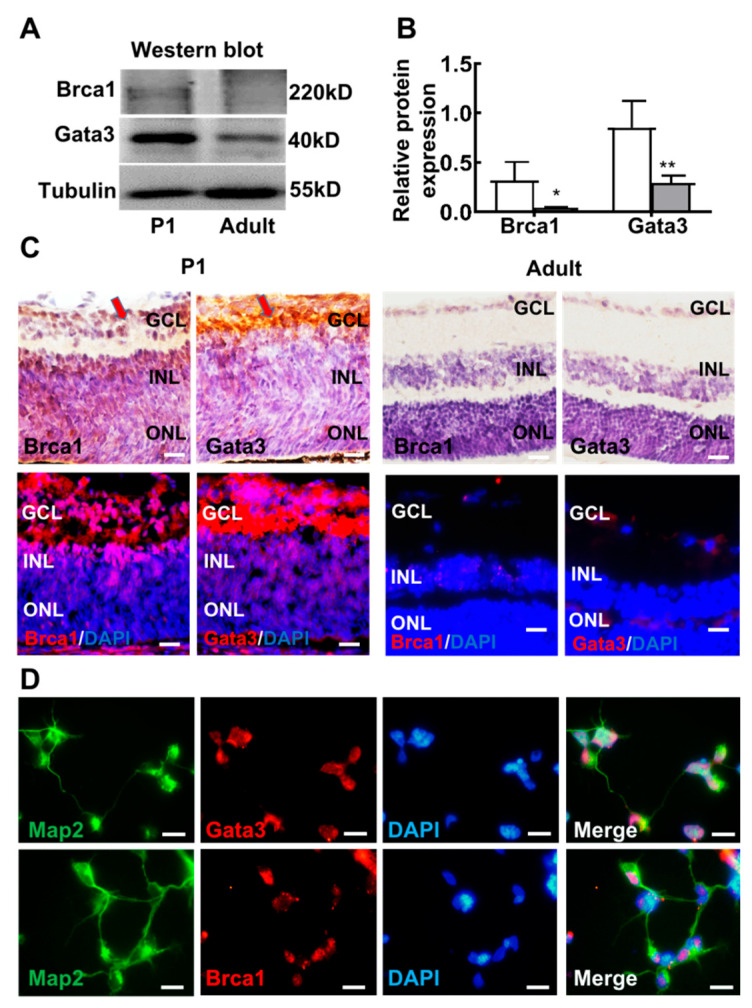
The expression profile of *Gata3* generally parallels that of *Brca1* in the retina. (**A**) Both *Gata3* and *Brca1* are developmentally downregulated in the mouse retina in vivo, as evidenced by Western blot. (**B**) The relative protein expression levels of *Gata3* and *Brca1* are presented as histograms. (**C**) The expression and localization profiles of *Gata3* are coincident with that of *Brca1* in the mouse retina, as evidenced by immunohistochemistry and immunofluorescence assays, *Gata3* and *Brca1* were stained as red in immunofluorescence. (**D**) *Map2* (green, cell cytoplasm) marks neurons, both *Brca1* (above, red, cell nuclei) and *Gata3* (below, red, cell nuclei) were expressed in P1 retinal neurons. Scale bars represent 20 μm. The asterisks indicate statistically significant differences (* *p* < 0.05, ** *p* < 0.01).

**Figure 7 ijms-23-13860-f007:**
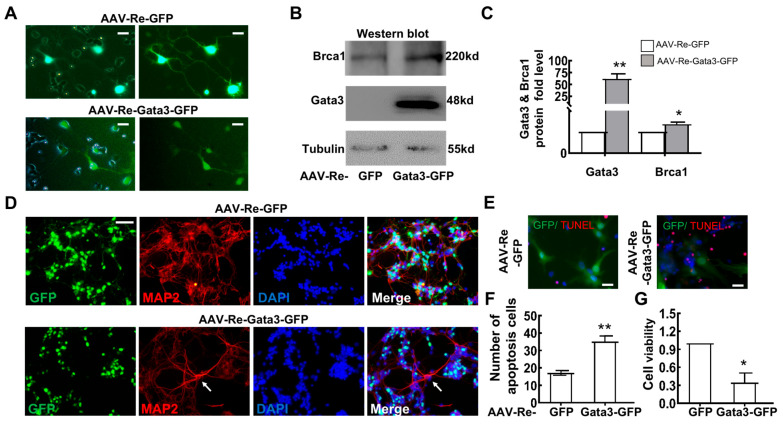
*Brca1* might be transcriptionally regulated by *Gata3* in vitro. (**A**) Primary retinal neurocytes (P1) were infected with AAV-Re-*Gata3*-GFP (green) or AAV-Re-GFP (green) adenoviruses. (**B**) AAV-Re-*Gata3*-GFP transfection upregulated the expression of not only *Gata3* but also *Brca1* in primary retinal neurocytes. (**C**) The relative protein expression levels of *Gata3* and *Brca1* are presented as histograms. (**D**) *Gata3* overexpression impeded differentiation in primary retinal neurocytes, as evidenced by staining with anti-*Map2* (red) in primary retinal neurocytes treated as above (green). (**E**) The number of apoptotic cells is presented as a histogram. (**F**) Exogenous *Gata3* significantly reduced the viability of primary retinal neurocytes. The asterisks indicate statistically significant differences (* *p* < 0.05, ** *p* < 0.01). Scale bars represent 20 μm.

## Data Availability

The datasets generated during and/or analyzed during the current study are not publicly available due to following study request, but are available from the corresponding author on reasonable request.

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
