# Peer review of "Brca1 Is Regulated by the Transcription Factor Gata3, and Its Silencing Promotes Neural Differentiation in Retinal Neurons"

_ijms, 2022, doi:10.3390/ijms232213860_

Round 1

Reviewer 1 Report

Major comments:

In Figure 1D, Brca1 silencing do not result in any change in Map2A/B expression, but in significant changes in Map2C/D. May the authors better explain the significance of Map2 isoforms during development/differentiation processes? Moreover, lowest molecular weight Map2C/D isoform seems to be specifically increased in Brca1 silenced cells, and this change is not so strong in the case of the highest molecular weight Map2C/D isoform. May the authors explain this data?

Why is viability reduced when Gata3 or Brca1 are silenced? It is because an increase in cell death, a decrease in proliferation or both processes may be involved?

Higher magnification microphotographs should be presented in Figure 3C to better show the nuclear colocalization of Gata3 and Brca1 (possibly including a red+green overlay).

 In the figure 6A western blot are performed on P1 and adult mouse retinas. Including embryonic tissues should be also interesting.

 In the figure 6B the expression of both Gata3 and Brca1 in the different retinal layers is not clear, probably because the hematoxylin counterstaining. For instance, Brac1 seems to be localized also in outer nuclear layer and outer plexiform layer. Maybe immunofluorescence images are more appropriate to provide a better overview of Gata3 and Brca1 expression in the retinal layers.

 Labeling also Brca1 in P1 retinal neurons would be interesting and could be included in the Figure 6C.

TUNEL staining and quantification should be included in the material and method section. Moreover, it is not clear to me if the number of apoptotic cells quantified in Figure 7E has been calculated by counting TUNEL/number of nuclei or TUNEL positive cells/field. Finally, TUNEL detects cell death (and not specifically apoptosis), then the sentences in which apoptosis is mentioned should be reformulated.

Primary retinal neurocyte is a very heterogeneous populations, composed by different neuronal types (e.g. precursors of ganglion cells, horizontal cells, bipolar cells, glial cells, etc.), having very different patterns of neurite outgrowth. For this reason, I think you should remove the data of neurite outgrowth presented in 7D or, alternatively, perform the same analysis after selecting a specific neuronal population (e.g. ganglion cells isolated by immunopanning).

In lines 418-420 you say: “Gata3/Brca1 is likely to be relevant for neuronal differentiation, and their abnormal expression attenuates neural physiological activity in adult retinal neurons.” But data about physiological activity in adult retinal neurons are not provided in this work.

Minor comments

Line 172-173: “Since Gata3 positively regulated the expression of Brca1, Gata3 may also be involved in neuron differentiation. To verify this hypothesis, the cells were double stained with anti-Nestin and anti-Map2”. Which cells?

Results in Figure 5A have been previously published by the same group (Pei Chen et al., 2022). This work should be at least cited.

Line 203: INL= inner nuclear layer

Reviewer 2 Report

The manuscript of Zuang et al describes the role of Brca1 and Gata3 in neuronal differentiation. The manuscript is well written but in this form, it is not suitable for publication.

1) The authors should add a double labelling Nestin-BRca1 and MAP2-Brca1 in figure 2. It is possible that all cells are not transfected and this immunofluorescence will permit to effectively reveal the morphology of siRNA neurons compared to not transfected cells.

2) the authors claim that Brca1 silencing reduces cell viability (rows 100-101). But if the cells are differentiated why not are they viable?

3) As before seen for Brca1, the authors should show double immunofluorescence for Nestin-Gata3 and MAP2-Gata3 in overexpressing and silencing Gata3 cells in figure 5.

4) As before, why does the Gata3 silencing determine the reduced viability? Not are variables the differentiated cells? (rows 180-181)

5)the authors should show immunofluorescence for Nestin and MAP2 in primary retinal cells infected with Gata3 to reveal the differentiated and undifferentiated cells 

Round 2

Reviewer 1 Report

Manuscript can be accepted in present form